# Small Bowel Capsule Endoscopy and Enteroscopy: A Shoulder-to-Shoulder Race

**DOI:** 10.3390/jcm12237328

**Published:** 2023-11-26

**Authors:** Ana-Maria Singeap, Catalin Sfarti, Horia Minea, Stefan Chiriac, Tudor Cuciureanu, Robert Nastasa, Carol Stanciu, Anca Trifan

**Affiliations:** 1Department of Gastroenterology, Faculty of Medicine, “Grigore T. Popa” University of Medicine and Pharmacy, 700115 Iasi, Romania; ana.singeap@umfiasi.ro (A.-M.S.); victor.sfarti@umfiasi.ro (C.S.); andrei-chiriac@umfiasi.ro (S.C.); cuciureanu.tudor@umfiasi.ro (T.C.); robert-radu_v_nastasa@d.umfiasi.ro (R.N.); carol.stanciu@umfiasi.ro (C.S.); anca.trifan@umfiasi.ro (A.T.); 2Institute of Gastroenterology and Hepatology, “St. Spiridon” University Hospital, 700111 Iasi, Romania

**Keywords:** small bowel, capsule endoscopy, device-assisted enteroscopy, diagnostic techniques, therapeutics, middle gastrointestinal bleeding, small bowel tumors, Crohn’s disease, artificial intelligence

## Abstract

Traditional methods have their limitations when it comes to unraveling the mysteries of the small bowel, an area historically seen as the “black box” of the gastrointestinal tract. This is where capsule endoscopy and enteroscopy have stepped in, offering a remarkable synergy that transcends the sum of their individual capabilities. From their introduction, small bowel capsule endoscopy and device-assisted enteroscopy have consistently evolved and improved, both on their own and interdependently. Each technique’s history may be told as a success story, and their interaction has revolutionized the approach to the small bowel. Both have advantages that could be ideally combined into a perfect technique: safe, non-invasive, and capable of examining the entire small bowel, taking biopsies, and applying therapeutical interventions. Until the realization of this perfect tool becomes a reality, the key for an optimal approach lies in the right selection of exploration method. In this article, we embark on a journey through the intertwined development of capsule endoscopy and enteroscopy, exploring the origins, technological advancements, clinical applications, and evolving inquiries that have continually reshaped the landscape of small bowel imaging.

## 1. Introduction

In the realm of modern medicine, the diagnosis and management of gastrointestinal disorders have been revolutionized by two remarkable techniques: capsule endoscopy (CE) and enteroscopy. These cutting-edge methods, sharing a unique synergy, not only complement each other but have also catalyzed a transformation in the way clinicians perceive and explore the enigmatic world within the human small bowel. Traditional diagnostic and imaging methods, while valuable, have their limitations when it comes to unraveling the mysteries of the small bowel, an area historically seen as the “black box” of gastroenterology. This is where CE and enteroscopy have contributed, providing a remarkable collaboration that exceeds the potential of each technique individually. 

With its miniature camera encased in a swallowable capsule, CE has brought a non-invasive revolution in the visualization of the gastrointestinal tract’s innermost reaches. On the other hand, enteroscopy, with its ability to biopsy and provide therapeutic interventions, has enabled practitioners to directly address the pathologies discovered during CE examinations. Enteroscopy itself has remarkably evolved, from traditional push-enteroscopy to sophisticated device-assisted techniques. Starting from 2001, the birth year of both CE and device-assisted enteroscopy, the interplay between these two techniques has provided an exquisite example of how innovation, collaboration and pursuit of knowledge can advance patient care.

This article explores the parallel evolution of CE and enteroscopy, encompassing their origins, technological advancements, clinical applications, and the ongoing research that continuously redefines the domain of small bowel imaging.

## 2. Small Bowel Exploration—Between Revelations and Unanswered Inquiries

Until the 2000s, small bowel exploration posed a significant challenge in the field of gastrointestinal tract diagnostics. The small bowel had long been a relatively inaccessible area for direct examination. Traditional diagnostic methods relied on imaging studies such as X-rays–barium swallow tests, upper endoscopy, and, introduced in the 1990s, push enteroscopy. A barium examination, performed either as a follow-through procedure or enteroclysis, allowing better intestinal distension, can assure diagnostic yields ranging from 6% [1] to no more than 21% for gastrointestinal bleeding [2], around 16% for suspected Crohn’s disease [3], and may reach 46% in the detection of small bowel tumors [4], although exposing patients to a radiation risk. Upper endoscopy comes with the limitation of not being able to see beyond the second duodenum. Push enteroscopy allows further small bowel examination, including the duodenum and proximal jejunum to approximately 50–100 cm beyond the Treitz ligament [5]; nonetheless, this technique is laborious, time-consuming, and carries a risk of complications such as perforation, bleeding, or pancreatitis [6]. Even if other newer exploration methods have brought progress, they still present some disadvantages for the current practice: entero-computed tomography and entero-magnetic resonance are not able to directly visualize the mucosal surface and, consequently, they have a relatively low sensitivity for the detection of the incipient lesions of Crohn’s disease [7]. Abdominal computed tomography angiography and marked red-cell scintigraphy, undoubtedly, shed light on the diagnosis of suspected small bowel bleeding, but the diagnostic success of both methods is determined by the activity and a minimum bleeding rate of 0.5 mL/min and 0.2–0.4 mL/min, respectively [8]. It is equally worth mentioning the limited accessibility of certain methods, especially magnetic resonance and nuclear techniques, which encompasses a significant drawback for routine clinical practice. 

Therefore, to address the need for improvement in small bowel pathology management, innovative tools had to be developed.

The first revolutionary solution, successfully fulfilling the goal of non-invasiveness, was ***small bowel capsule endoscopy***. The first-generation capsule endoscopy, known as the M2A capsule, was launched in 2001, about the size of a large pill, and had a small camera, light source, and transmitter [9]. Its appearance, purely and simply, revolutionized gastrointestinal tract exploration, allowing non-invasive imaging of the small intestine. Only two years after its entrance into clinical practice, small bowel CE had already been used in more than 4000 patients [10]. Subsequent generations of capsule endoscopes featured improved image quality and better maneuverability, becoming smaller and easier to swallow. By the late 2010s, several companies were manufacturing capsule endoscopy systems, offering various features and enhancements: increased field of view, advanced optics with better image resolution, and improved battery life, as well as software progresses, allowing optimized images analysis, automatic detection of identical images, suspected blood indicators, and incorporated scores [11]. Today, after more than twenty years of utilization, millions of capsule examinations have been performed worldwide, while, due to its safety, patient comfort, and the possibility of visualizing the entire small bowel mucosa, it has become the first-line examination in most small bowel pathologies [12]. Nevertheless, specific precautions have to be considered before undergoing a small bowel CE examination. Avoiding the procedure is recommended for individuals with suspected or known stenosis of the gastrointestinal tract, unless intestinal patency is confirmed. In cases of swallowing or gastric motility disorders, the option of endoscopically placing the capsule may be considered [13]. There is currently a lack of data regarding the safety of CE in pregnant women. Implanted cardiac devices like pacemakers or cardioverters were initially considered a contraindication, while subsequent studies have dispelled concerns about potential dysfunction in either the capsule or cardiac devices [14]; even so, manufacturers still maintain this contraindication. The use of a remote control for insulin infusion pumps has been suggested to potentially interfere with the CE transmission [15]. 

However, despite its successes, CE has some limitations, such as a lack of real-time control; the limited battery life, which may result in an incomplete examination; and the inability to take biopsies or perform therapeutic interventions. Currently, efforts are being made to address these limitations by developing smart capsules that can actively move within the gastrointestinal tract, obtain biopsy samples, or even treat lesions [16,17]. Nevertheless, until such progress is realized, small bowel CE will require complementary techniques like enteroscopy, to ensure comprehensive evaluation, diagnosis, and the right therapeutic strategy.

The advent of CE has had, as a direct consequence, a growing requirement for a proper endoscopic technique to directly access the small bowel and allow lesions biopsy and targeted therapeutic interventions. Thus, various developments have been made and several techniques of ***device-assisted enteroscopy (DAE***) have become available in practice in the last twenty years, allowing the application of many types of endotherapy, such as polypectomy, endoscopic mucosal resection, hemostasis, stricture dilation, foreign bodies retrieval, and jejunostomy placement [18].

*The double balloon enteroscopy (DBE)* system, including an enteroscope, an overtube, and two inflatable balloons—one at the distal end of the endoscope and the other attached to the overtube—works using the “pull-and-push” technique; the distal balloon helps anchor the endoscope in place, preventing it form slipping out, while the proximal balloon helps advance the endoscope deeper into the small bowel [19]. Two methods of insertion are possible, either antegrade—via the mouth, passing the esophagus and the stomach first—or retrograde, via the anus and colon [20]. The antegrade approach usually ensures a length of insertion between 230 and 360 cm, while for the retrograde approach, there is an estimated length of examined small bowel of 100–130 cm [21]. After the first decade of use, a systematic review, comprising 66 original English-language articles and more than 12 thousand procedures, showed a pooled detection rate for all indications of 68.1% and a pooled total enteroscopy rate of 44%, using an antegrade-only or combined approach [22]. More recent studies have shown higher diagnostic yields, reaching 75–78.7% according to two Korean retrospective studies [23,24]. The huge advantages of DBE consist in its ability to take a biopsy and perform therapeutic techniques, such polypectomy, argon plasma coagulation, stent placement, dilatation, and foreign body removal. The overall complication rate of DBE is around 3.5%, including perforation, bleeding, and pancreatitis; their frequency varies according to the type of DBE, being lower in diagnostic DBE (under 1%) and higher (4.3%) in therapeutic procedures [25]. An increase in serum amylase and lipase was reported in almost half of patients [26,27], while postprocedural acute pancreatitis, especially reported after antegrade insertion, had a frequency of 0.3% and was considered to be the consequence of mechanical maneuvers during the push-and-pull movements [28]. Other drawbacks of DBE involve a steep learning curve, being time consuming, and the need for sedating patients.

*The single-balloon enteroscopy (SBE)* technique was designed to simplify the push-and-pull method and relies on the use of a single balloon attached to the tip of the overtube, resulting in a reduced setup time and more user-friendly balloon control panel [29]. There is a 15–25% rate of complete enteroscopy, lower compared to DBE [30,31], and the reported depth of insertion is between 133 and 270 cm beyond the Treitz ligament for the antegrade approach and 73 and 199 for retrograde intubation [32]. The reported overall diagnostic yields of SBE ranged from 41% to 65% [33,34]. Notably, the diagnostic yield increased if SBE was performed after a previously positive CE, reaching almost 80% [35].

*The spiral enteroscopy (SE),* introduced into medical practice in 2007, is a high-performance examination of the small intestine by means of rotated-to-advance technology that uses a tube with a helical appearance (discovery small bowel), fixed on an endoscope to keep it in a stable position. This technique involves the participation of two operators, who manually apply discrete rotary movements to the tube to fold the small intestine over the enteroscope, which improves movement during its insertion and withdrawal [36,37]. The antegrade approach of spiral enteroscopy has been frequently used in various studies, with an average depth of intubation that varied between 200 and 346 cm. The main benefit is related to the substantial reduction in investigation time, obtaining diagnostic and therapeutic results comparable to DBE for a similar maximum insertion depth [38,39]. However, the success rate of this technique did not exceed 10%, mainly due to the risks associated with the retrograde passage of the enteroscope [40]. Akerman et al. reported a frequency of only 0.3% for major complications associated with the application of this method of examining the small intestine in a group that included 2950 patients; eight cases of perforation were diagnosed, but no patients presented with acute pancreatitis [37]. This demonstrates that SE presents a minor risk for the generation of this pancreatic pathology compared to DBE and SBE. Numerous studies have suggested that an endoscopist can quickly gain competence in SE if they are trained and perform at least five procedures [41,42]. More recently introduced into practice, the *motorized spiral enteroscopy* (PowerSpiral Enteroscopy, Olympus Medical Systems Corporation, Tokyo, Japan) appeared in 2016 as a promising method replacing SE [43]. This equipment consists of a reusable endoscope (technical specifications: length—168 cm, outer diameter—12.8 mm, inner channel diameter—3.2 mm), to which a short spiral-like tube is attached. The two components are connected to an integrated motor that induces rotational movements by pressing a pedal provided with a visual force indicator [43,44]. However, unfortunately, despite expectations, several major adverse associated events were noted, including deaths, and the method was withdrawn by the manufacturing company [45]. 

In the same period, a new *balloon-guided endoscopy (BGE)* procedure appeared, known as NaviAid System, developed by Smart Medical Systems in Ra’anana, Israel, whereby a dedicated through-the-scope balloon is inserted into the endoscope or colonoscope working channel [46]. Additionally, a catheter can be inserted through the working channel of the endoscope, creating a double-balloon approach. Through the connection of these components, an anchoring device is formed that can be progressively advanced through repeated pushing and pulling maneuvers in the lumen of the small intestine, having the role of reducing the procedure time necessary to perform a deep enteroscopy. In the opinion of various specialists, it was considered that balloon-guided endoscopy (BGE) can be applied for both the antegrade and retrograde paths whenever needed, being considered an “on-demand” procedure [47,48]. Moreover, in the case of therapeutic procedures, the balloon catheter can be removed and reinserted, if necessary, to complete the procedure [48]. Although current information is relatively limited, it is suggested for the anterograde approach of this technique, it is suggested to observe a maximum insertion depth limit of approximately 120 cm, while for the retrograde approach, this should not exceed 110 cm [49]. 

Quality indicators have been a major preoccupation for all gastrointestinal procedures; DAE has generally assimilated the desiderates of other invasive direct procedures, while requirements adapted for SBCE have also been formulated [50,51,52]. 

Simultaneously facing the advantages and drawbacks of small bowel CE and enteroscopy (Table 1) helps clinicians understand the urgent requirements and to design the profile of an ideal tool for small bowel exploration: non-invasive, able to accurately visualize the entire small bowel mucosa, allowing biopsies, and capable of therapeutic interventions.

Until such an ideal technique becomes a reality, the key for an optimal practice remains the selection of the appropriate diagnostic approach (be it CE, enteroscopy, or both), in such manner that the advantages exceed the limitations in every clinical case scenario.

## 3. The Synergy of Capsule Endoscopy and Enteroscopy in Various Clinical Settings

The main indications for small bowel exploration are suspected small bowel bleeding, unexplained iron deficiency anemia, Crohn’s disease, and the suspicion of small bowel tumors; there are other less frequent indications, such as celiac disease and polyposis syndromes.

*Suspected small bowel bleeding* is the most common indication for both small bowel CE and enteroscopy; actually, when the bleeding source is thought to be in the small bowel, from beyond the ampulla of Vater to the ileo-cecal valve, the term middle gastrointestinal bleeding (MGIB) should be used [53]. Present guidelines recommend small-bowel CE as first-line examination in suspected MGIB, due to its capability to entirely visualize the small bowel mucosa and owing to its safety profile, as soon as possible after the bleeding episode [12]. Compared to SBCE, DAE has therapeutic advantages, but it is invasive and has a lower rate of complete examination. However, the DY of DBE is significantly improved if performed after a positive SBCE examination, going from 56% to 75% (odds ratio 1.79, 95%CI 1.09–2.96; *p* = 0.02), as shown by a comprehensive meta-analysis [54]. SBCE should be performed as the first examination if available, except in cases of massive, active bleeding, where angio-computed tomography or angiography should be performed urgently [12]. If SBCE is not available, contraindicated, or in other selected cases, based on local expertise, DAE may be performed as a first-line examination, with the optimal time being the same interval of 48 h, and a maximum 72 h, after the bleeding episode. Several studies have shown that diagnostic yield (DY) is improved when small bowel exploration is performed, either using SBCE or DAE, as soon as possible [55,56]. A recent large meta-analysis including 39 studies and 4825 patients showed higher pooled DY for early SBCE, being 83.4%, 81.3%, and 63.6% for SBCE performed in the first 24, 48 and 72 h, respectively [57]. If available, DAE is favored when the clinical suspicion orientates towards the need for a therapeutic treatment, due to its intervention potential. May et al. [21] reported that DBE procedures were performed in 60% of 353 patients who were admitted due to bleeding, and DBE was found to be diagnostically and therapeutically beneficial in 75% and 67% of these cases, respectively. A sequential approach is normally preferred, with both SBCE and DBE performed within several days of each other; the chances of a positive diagnosis and a successful therapeutic intervention are greater when the interval between the bleeding episode and the first procedure, and also between the two procedures, is shorter [57]. SBCE may guide the route of insertion of DAE, the oral or anal approach being chosen according to the localization of the lesion, as described by SBCE. So far, there are no absolute indicators for lesion localization during SBCE procedure; however, the available time-based indices and CE progress indicator tool accurately determined the segment and correctly indicated the right insertion route for DAE, with a high success rate ranging from 78% to 100%, as shown by a recent systematic review [58]. Therefore, considering their distinctive advantages, capabilities, and limitations, SBCE is unanimously recommended as the first-line examination in suspected MGIB, followed by DAE for lesion confirmation and therapeutic intervention.

In *iron deficiency anemia (IDA),* small-bowel exploration must be considered after a comprehensive prior work-up. After a negative upper endoscopy with negative gastric and duodenal biopsies, and after negative ileocolonoscopy, SBCE is indicated [18,59]. There has been no consensus so far regarding the role of fecal occult blood testing as a filter for small bowel exploration. Studies have proved that SBCE has a high DY—especially in male and old patients with low hemoglobin values or high transfusion requirements—and an excellent safety profile [60,61,62]. Moreover, if negative, there is evidence in favor of a very low rebleeding risk [63]. There have been few studies regarding the DY of DAE exclusively in unexplained IDA, but the performance parameters in terms of diagnostic and therapeutic interventions are considered similar to those reported for suspected MGIB. 

In *suspected Crohn’s disease* (CD)**,** if ileocolonoscopy is negative, SBCE is recommended as first-line investigation in patients, after retention risk has been excluded.

Thanks to its ability to non-invasively visualize the entire small-bowel mucosa, SBCE is preferred as the first-line examination. DY was proved superior to small-bowel follow through, enteroclysis, push-enteroscopy, and generally similar to computed tomography enterography and magnetic resonance enterography [64,65]. However, there are studies showing that, for incipient lesions and for the proximal small-bowel, SBCE appears superior to cross-sectional imaging [66,67]. Nonetheless, a major drawback of SBCE is the inability to take biopsies, which is why all features described by SBCE must be cautiously interpreted. Mow’s criteria, widely used but not validated, suggest that more than three ulcerations are diagnostic for CD, in the absence of nonsteroidal anti-inflammatory drugs in the previous four weeks [68]. In patients with suspected CD, given that there is no gold standard for diagnosis, and especially when there are not enough other positive elements for sustaining it, a histological confirmation by enteroscopy is generally needed. Taking biopsies will help differentiate from other small bowel diseases presenting mucosal ulcerations, such as infections, eosinophilic enteritis, or neoplasia. 

The retention rate of SBCE in suspected CD is 2.35%, as shown by a recent meta-analysis [69]. When there are obstructive symptoms, dedicated small-bowel cross-sectional imaging methods must be used first or a patency capsule may be administrated before SBCE. In cases where a capsule is retained and medical intervention is not successful, retrieval can be attempted through DAE, thereby avoiding surgery; in certain cases, simultaneous endoscopic balloon dilation may be also performed [70]. 

In *established Crohn’s disease*, SBCE is useful for assessing the disease extension, the disease activity, and the response to therapy, as well as the postoperative recurrence [71,72]. The Lewis score and the Capsule Endoscopy Crohn’s Disease Activity Index (CECDAI) are reciprocally well correlated regarding disease activity, but they both appear to be poorly correlated with clinical and laboratory parameters [73,74]. However, mucosal inflammation, as assessed by capsule scores, has a prediction role for poor outcome, even if there is clinical and biochemical remission at the time of examination [75]. Thus, small bowel examination by SBCE is indicated whenever there is confidence that its result could change the management. A novel panenteric capsule endoscopy was recently developed, allowing a single assessment of both small and large bowel mucosal inflammation [76]. A European multicenter observational cohort study demonstrated its feasibility in daily clinical practice, permitting an accurate estimation of the prognosis and guide intensification of the therapy [77]. 

The retention risk for SBCE is considerably higher in established CD, reaching 8.2%, as shown by a meta-analysis [78]. Capsule retention might be avoided by thorough prior imaging or by using a patency capsule. When capsule retention occurs, medical treatment is initially attempted, followed by DAE, if necessary [18], unless there is clinically manifest acute intestinal obstruction needing immediate surgery.

*Suspected small bowel tumors (SBTs)* benefit from SBCE as a first-line investigation. Since the introduction into the clinical practice of the SBCE, the diagnostic rate of SBTs has doubled [56], and most SBTs are found when SBCE is performed for suspected MGIB or unexplained IDA. SBTs accounts for approximately 5% of causes of unexplained gastrointestinal bleeding or IDA [79]. SBCE has an increased DY for SBTs compared to push enteroscopy [80] but similar detection rates when compared to DAE, both in the case of SBE [81] and DBE [82]. However, for several subgroup of patients, there is a demonstrated risk of false-negative SBCE results, such as for submucosal mass with minimal luminal expression or for those lesions proximally located in the small bowel [83]. For these reasons, when SBTs are suspected, cross-sectional small bowel imaging must follow a negative examination. Nonetheless, recent studies showed that DBE identified the small bowel neuroendocrine neoplasms with better accuracy than computed tomography, magnetic resonance-enterography, and somatostatin receptor imaging [84]. For cases where the SBCE encounters the dilemma of differentiating a real mass from a bulge, DAE or cross-sectional imaging can help. 

Undoubtedly, when histological confirmation is essential, biopsies are needed and enteroscopy emerges as an alternative or complementary method. DAE is preferred as a first-line method over SBCE when imaging tests have already described the SBT. Even if SBCE may provide suggestive macroscopic features for a certain type of SBT, it remains a visual technique, sometimes with indefinite SBT localization data, and further assessment using DAE for biopsies or tattoo placement might be necessary before surgery [85].

*Peutz–Jeghers syndrome* is another clinical setting where SBCE and DAE are successfully used as complementary methods. The studies so far have shown a high concordance in the identification of small bowel polyps between SBCE and DAE, with comparable DY for polyps of more than 15 mm diameter [86]. Surveillance of Peutz–Jeghers syndrome involves SBCE, followed if needed by DAE with polypectomy, as a strategy to prevent polyp-related complications and manage premalignant conditions.

In *refractory* or *non-responsive celiac disease,* SBCE is indicated to identify complications such as ulcerative jejuno-ileitis, enteropathy-associated T-cell lymphoma, or small-bowel adenocarcinoma. Due to a sequential approach—upper endoscopy followed by SBCE and then DAE for tissue sampling—the overall DY of these subtypes of celiac disease has increased remarkably. SBCE showed a DY of nearly 50% in patients with refractory celiac disease, identifying SBTs in up to 10% of cases [87]. When complicated celiac disease is suspected, studies have suggested that SBCE should be the first-line approach, in order to detect the cases that need further histological assessment using DAE, with around 40% of cases being unreachable by conventional upper endoscopy [88].

## 4. Further Challenges and Goals for the Future

Since their debut, SBCE and DAE have undergone continuous evolution.

SBCE has registered remarkable technological and software advances. Today, the available SBCE devices are able to acquire high-resolution images and have an adaptive frame rate, an extended life battery, and post-processing chromoendoscopy facilities [89].

Undoubtedly, SBCE technology was pushed further by the advent of DAE. Magnetically-guided capsule systems are now in use for combined gastric and small bowel examination [90], and panenteric capsules have been developed [76].

However, there are some unresolved issues, such as the exact localization of lesions described by SBCE, the large number of images requiring a long interpretation time, the lack of capabilities for taking biopsies, and the inability to perform therapeutic treatments.

Further progress is needed regarding tridimensional localization facilities and tracking systems, in order to accurately determine the right insertion route for DBE. 

Some potentially revolutionary techniques are in the initial stages of development, hopefully in the near future allowing interventions such as tissue sampling, drug delivery, and local coagulation [18].

In the future, the implementation of artificial intelligence (AI) might solve the triage of normal images or even the characterization of abnormalities. Deep learning models have been suggested for blood identification, demonstrating a significantly elevated sensitivity, specificity, and precision, exceeding 98% [91,92]. However, this recognition primarily relied on static images, lacking validation on video segments. Additionally, the models did not provide blood feature details such as aspect (fresh, melena, or clots) or quantity. The application of AI using a convolutional neural network (CNN) exhibited automatic detection of angioectasias with remarkable sensitivity (98.8%) and specificity (98.4%) [93]. Nonetheless, further prospective clinical studies are imperative to validate these promising outcomes. The diverse range of ulcerative lesions, encompassing variations in size, shape, and cause, present a significant challenge for detecting erosions and ulcers. The identification of small bowel ulcers through deep learning models utilizing CNN and support-vector machines (SVM) demonstrated an approximately 88% sensitivity and 91% specificity [94,95], with a shorter reading time. A deep-neural-network-based system achieved automatic detection of SBTs with a sensitivity of 91% and specificity of 80%, as demonstrated by Saito et al. in 2020 [96]. The learning process relied on static images and, at present, there is a lack of extensive studies involving video data. A significant challenge persists in distinguishing genuine tumors from “fake” ones or normal, non-significant structures, such as bulges, lymphangiectasias, lymphoid nodular hyperplasia, Brunner glands, phlebectasias, normal folds, or even the papilla. In fact, this remains an ongoing dilemma for human capsule endoscopy readers, and further data and algorithms are eagerly awaited to address this issue. AI has been also involved in assessing the quality of bowel preparation, with neural networks achieving over 95% accuracy [97]. 

However, despite the advancements and accomplishments to date, numerous gaps and uncertainties persist regarding the performance of AI. Until AI is thoroughly validated and deemed entirely trustworthy, the responsibility for interpreting images will continue to rest with humans, specifically a physician possessing training and expertise in capsule endoscopy reading. 

Finally, SBCE might raise concerns in the era of “green endoscopy” [98]; the number of SBCE examinations has continuously increased since its introduction, a fact that brings into discussion the issue of the millions of batteries and non-degradable components arriving into the environment. 

DAE has also seen progress over time. Having on its side the incontestable advantage of providing tissue sampling and interventional techniques, parallel efforts have been made over time in two directions: insertion depth, and patient safety. The goal of maximum length small bowel intubation was probably inspired by the ability of SBCE to entirely visualize the small bowel mucosa. Fluoroscopy proved useful in order to estimate the depth of insertion but also for scope positioning in patients with surgically-altered anatomy and in therapeutic interventions such as enteral-tube placement, stricture dilation, and endoscopic retrograde cholangiopancreatography [18]. Concerning the safety of the method, complication rates have been a persistent concern; when needed, recall measures are applied, as for motorized enteroscopy [45]. 

The anticipate progress in deep learning is expected to eventually translate into DAE also, ensuring optimized lesion detection. Moreover, the integration of robotics into flexible endoscopy seems to offer increased stability and precision for complex therapeutic procedures, potentially extending its advantages towards deep enteroscopy [99]. 

Together with these progresses, a refinement in quality indicators is expected to be formulated, in terms of indications, performance, safety, terminology, and impact on patient management.

## 5. Conclusions

From their introduction, SBCE and DAE have consistently evolved and improved over time, both on their own and interdependently. As they collaborate, achieving excellence and success are equally crucial. Each technique’s history may be narrated as a tale of accomplishment, while their interplay and interaction has revolutionized the small bowel approach.

Both SBCE and DAE have advantages that could ideally be summed together into the perfect technique: safe, non-invasive, and able to examine the entire small bowel, take biopsies, and apply therapeutical interventions. 

Until the realization of this perfect tool becomes a reality, the key for an optimal approach lies in the right selection of exploration method. Whether this is SBCE, DAE, or a combination of both, each clinical setting must be managed so that the benefits outweigh the limitations.

## Figures and Tables

**Table 1 jcm-12-07328-t001:** The advantages and limitations of small bowel capsule endoscopy and enteroscopy.

Criteria	Small Bowel Capsule Endoscopy	Enteroscopy
Procedure type	Non-invasive	Invasive
Patient comfort	Well-tolerated and patient-friendly	May require sedation, potentially causing discomfort
Visualization	Enables panenteric mucosal visualization; however, limited by the battery life The application of artificial intelligence has shown potential for enhanced image analysis and interpretation	Facilitates direct, real-time visualization of the small bowel mucosa; however, visualization length is limited
Maneuverability	Limited control over the capsule’s movement	Provides control and maneuverability, enables targeted examinations
Diagnostic yield	Demonstrates superior diagnostic efficacy for mucosal lesions, particularly in suspected small bowel bleeding No definitive diagnosis, may necessitate biopsy	Exhibits enhanced diagnostic sensitivity for lesions located in the proximal segments of the small bowel, allows tissue sampling
Radiation exposure	No radiation exposure	May involve exposure to radiation during fluoroscopy
Mutual support	Guides the insertion route for enteroscopy	Targets lesions seen in capsule endoscopy
Procedure time	Shorter procedure duration; time needed for capsule ingestion and subsequent image reading	Lengthier procedural timeframe
Therapeutic interventions	Primarily a diagnostic modality, without the ability for therapeutic interventions	Facilitates therapeutic maneuvers, including biopsies, polypectomies, and hemostasis, contributing to both diagnosis and treatment
Main contraindications and precautions	Gastrointestinal tract obstruction, swallowing disorders	May pose challenges in patients with strictures or significant comorbidities
Accessibility	Easier to perform, dedicated training required	Requires specialized training and expertise
Complications	Rare complications, as the capsule is naturally excreted	Risk of complications such as perforation and bleeding
Cost	Involves costs associated with the capsule and related equipment	Entails increased overall expenses, attributed to specialized equipment, personnel, and facility requirements
Objectives and aspirations	Accuracy Biopsies Treatment	Effectiveness Safety

## Data Availability

Not applicable.

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
