# Peer review of "Small Bowel Capsule Endoscopy and Enteroscopy: A Shoulder-to-Shoulder Race"

_jcm, 2023, doi:10.3390/jcm12237328_

Round 1

Reviewer 1 Report

Comments and Suggestions for Authors

I read this review with interest. It is well written however, similar review were seen. I recommend making table or figure to explain the advantage and disadvantage of capsule endoscopy. 

Comments on the Quality of English Language

I do not have any comment.

Reviewer 2 Report

Comments and Suggestions for Authors

Thank you for the opportunity to review this manuscript by Singeap et al. I have the following comments for the authors.

1.     The authors strike a casual tone in the manuscript with the use of terms “racing and winning” that can confuse readers or make the topic seem less serious than it is. The “dark zone” is not a commonly used term to describe small bowel.

2.     The abstract and the introduction are repetitive with several similar phrases such as “synergy” and “intertwined evolution”

3.     The authors use the term “obscure gastrointestinal bleeding” which is an antiquated term. Guidelines (Gerson et al. Am J Gastroenterol 2015 PMID: 26303132) have suggested a shift from obscure gastrointestinal bleeding to suspected small bowel bleeding. 

4.     The authors should also include limited battery life with a potential for incomplete examination.

5.     The authors note that a retrograde approach for device-assisted enteroscopy achieves a depth of intubation of 100-130 cm.  The authors may want to clarify whether this depth of intubation related to small bowel length or including colon length. 

6.     The authors may want to include figures or tables that summarize the advantages and disadvantages of video capsule endoscopy and device-assisted enteroscopy to better help readers better understand the principles underpinning endoscopic evaluation of the small bowel

7.     On page 6, the authors use % for the 95% confidence interval for an odds ratio. It is important to note that 1.09-2.96 (the 95% confidence interval) should not have a % sign as the confidence interval is an odds ratio, not a percent.

8.     The authors may want to consider inclusion of a more updated manuscript that discusses quality indicators for capsule endoscopy and deep enteroscopy (Leighton et al. Am J Gastroenterol 2022 PMID: 36155365).

Comments on the Quality of English Language

The authors should have a native English speaker review their manuscript and suggest grammatical changes as they use terms such as “consensually” that have certain connotations that may distract from the substance of the manuscript.

Reviewer 3 Report

Comments and Suggestions for Authors

An interesting review.

The following are suggested to enhance the manuscript:

1. The data for each modality should be represented in a tabular way for easy review and comprehension 

2. The following may be referenced and discussed:

a. Pennazio M, Rondonotti E, Despott EJ, Dray X, Keuchel M, Moreels T, Sanders DS, Spada C, Carretero C, Cortegoso Valdivia P, Elli L, Fuccio L, Gonzalez Suarez B, Koulaouzidis A, Kunovsky L, McNamara D, Neumann H, Perez-Cuadrado-Martinez E, Perez-Cuadrado-Robles E, Piccirelli S, Rosa B, Saurin JC, Sidhu R, Tacheci I, Vlachou E, Triantafyllou K. Small-bowel capsule endoscopy and device-assisted enteroscopy for diagnosis and treatment of small-bowel disorders: European Society of Gastrointestinal Endoscopy (ESGE) Guideline - Update 2022. Endoscopy. 2023 Jan;55(1):58-95. doi: 10.1055/a-1973-3796. Epub 2022 Nov 24. PMID: 36423618.

Reviewer 4 Report

Comments and Suggestions for Authors

Dear authors, i read with intereet uour paper which clarly su.msrizes the topic described in the title. The paper is written in a teader friendly manner and is an up to dated report

Best regards

Reviewer 5 Report

Comments and Suggestions for Authors

The manuscript is well-written. However, it lack novelty. Authors should include information thatsould make a manuscript original and eye- catching.

They should also include figure and table.

Percations for capsule endoscopy should also be included.

Round 2

Reviewer 1 Report

Comments and Suggestions for Authors

Authors made efforts to improve the manuscript. I do not have any comment. 

Comments on the Quality of English Language

I do not have any comment. 

Reviewer 5 Report

Comments and Suggestions for Authors

I am glad to see that the authors have heard my opinion and suggestions. The manuscript has deserved to be published.